# Hot-Pressing Furnace Current Monitoring and Predictive Maintenance System in Aerospace Applications

**DOI:** 10.3390/s23042230

**Published:** 2023-02-16

**Authors:** Hong-Ming Chen, Jia-Hao Zhang, Yu-Chieh Wang, Hsiang-Ching Chang, Jen-Kai King, Chao-Tung Yang

**Affiliations:** 1Department of Applied Mathematics, Tunghai University, No. 1727, Sec. 4, Taiwan Boulevard, Taichung City 407224, Taiwan; 2Department of Computer Science, Tunghai University, No. 1727, Sec. 4, Taiwan Boulevard, Taichung City 407224, Taiwan; 3Innovation R&D Center, The Aerospace Industrial Development Corporation, No. 1-1, Hanxiang Rd., Xitun Dist., Taichung City 407803, Taiwan; 4Research Center for Smart Sustainable Circular Economy, Tunghai University, No. 1727, Sec. 4, Taiwan Boulevard, Taichung City 407224, Taiwan

**Keywords:** predictive maintenance, current monitoring, machine learning, monitoring algorithm, aerospace

## Abstract

This research combines the application of artificial intelligence in the production equipment fault monitoring of aerospace components. It detects three-phase current abnormalities in large hot-pressing furnaces through smart meters and provides early preventive maintenance. Different anomalies are classified, and a suitable monitoring process algorithm is proposed to improve the overall monitoring quality, accuracy, and stability by applying AI. We also designed a system to present the heater’s power consumption and the hot-pressing furnace’s fan and visualize the process. Combining artificial intelligence with the experience and technology of professional technicians and researchers to detect and proactively grasp the health of the hot-pressing furnace equipment improves the shortcomings of previous expert systems, achieves long-term stability, and reduces costs. The complete algorithm introduces a model corresponding to the actual production environment, with the best model result being XGBoost with an accuracy of 0.97.

## 1. Introduction

This research mainly uses artificial intelligence to monitor the equipment’s maintenance time and the equipment’s health, adjust the production schedule and purchase maintenance parts in advance, and reduce the production costs caused by downtime.

Fiber-reinforced polymer composites have recently been used as primary structural materials for aerospace vehicles. The manufacturing process of composite materials has adopted the process of intelligent manufacturing, including machine learning (ML) and artificial intelligence (AI).

The manufacturing process of composite materials equipment of a hot-pressing furnace requires at least eight hours or more. The cost of manufacturing parts, electricity, and labor required is more than USD 100,000 each time. The equipment maintenance of the hot-pressing furnace is required for abnormal monitoring and predictive maintenance. If an abnormal situation occurs in each device, it must be tested by maintenance staff, which is very time-consuming and inefficient. In the event of an anomaly, the loss can exceed USD 100,000 in waste. Therefore, it is hoped that the existing early warning monitoring system can reduce the need for warranty testing [1].

In the past, the expert system [2,3] was used as a monitoring process. However, many anomalies and emergencies remained uncorrected. Therefore, expert systems do not solve the problem in the long run. This research makes use of artificial intelligence. The method is presented. Through current monitoring, the power consumption of heaters and fans during the formation and transformation of the hot-pressing furnace is recorded online. We actively grasp the health of the hot-pressing furnace equipment to improve the lack of expert systems in the past, achieve long-term stable use, and reduce costs. Sacco et al. [4] used ML vision systems to develop machine learning applications in composites manufacturing for the University of South Carolina’s automated fiber placement process. Various sensors are used in industrial fields, including pressure, temperature, force, and vibration, but there is no complete process of automated data collection and analysis. Fault identification has become more important in the maintenance of induction motors. So, the condition of the bearing in induction machines must be continuously monitored. Because the amount of current data is huge, we argue that current anomaly detection requires a scalable system architecture ability including real-time data processing, resource efficiency, fault tolerance, and extensibility. Xiang et al. [5] identified abnormal samples in distribution transformer inspection reports, assisted inspectors to complete inspection work correctly and efficiently. Balouji et al. [6] integrated an advanced generative adversarial network to detect anomalies in FDIA data. Many scholars have proposed various artificial intelligence or machine learning methods to improve special machines or imaging of three-phase current [7,8,9]. Related methods of artificial intelligence and machine learning models [10] use machine learning and deep learning models, such as an SVR (Support Vector Regression) model, a KNN (K-Nearest Neighbor) model, and a radial basis function (RBF) neural network model. Other methods were also proposed: Hadi Salih et al. [11] proposed using the Discrete Cosine Transform (DCT) to analyze the speed and the Probabilistic Neural Network (PNN) to identify bearing failures.

An anomaly detection method based on local anomaly factors is proposed through experiments on real datasets, and the results were deemed accurate by [12]. Zeng et al. used the parallel symmetric multiprocessor computing machine for problems with high-dimensional training sets. This new technique has significant advantages in speed without reducing the generalization performance of the support vector machine (SVM) [13]. Wang et al. improved the effectiveness of a real power system experimentally with parameters optimization particle swarm optimization (PSO) algorithms [14]. The industry example in the current case [15] analyzes power consumption data for fault detection, and predictive maintenance, and provides the implementation code on the Web. Recently, industrial power systems [16,17,18,19] deployed the power consumption of 16 servers in this pilot implementation for a medium-sized enterprise. The system uses the Hadoop, Hive data warehouse with Spark. Data storage is applied in electric power big data and provides query functions for statistical and data performance tests. Extended deployment to intelligent cloud edge computing architecture, providing ML and deep learning implementation in the cloud edge environment, was realized. Zhu et al. combined Active learning (AL) and transfer learning (TL) for remaining useful life (RUL) prediction to design a method that is more practical with a lesser demand on the run-to-failure data under limited labeled samples and even no labeled samples [20].

The paper is arranged into four different sections. Section 1 is the introduction, Section 2 presents monitoring with artificial intelligence and algorithm description. Section 3 presents the current monitoring system and numerical results, and Section 4 is the conclusion.

## 2. Monitoring with Artificial Intelligence and Algorithm Description

This section proposes algorithms for monitoring current and equipment status and presents a tree diagram and mapping algorithm. It is an algorithm that is easy to design and apply to other machinery fault detection with high accuracy and has been introduced in [21,22].

Our detection status (Figure 1) for current abnormalities is divided into normal Algorithm $A_{1}$ and abnormal Algorithm $A_{2}$. In the case that $A_{2}$ is divided into Algorithm $A_{21}$ and Algorithm $A_{22}$, Algorithm $A_{21}$ is further subdivided into two types which are abnormal in Case 1 but do not affect the status and new abnormalities in Case 2 but can be classified. In the case of Algorithm $A_{22}$, it is hoped that there are no rules to find a method through deep learning or different mechanical learning, and it can be classified back to $A_{21}$. Our detection status (Figure 1) for current abnormalities is divided into normal Algorithm $A_{1}$ and abnormal Algorithm $A_{2}$. In the case that $A_{2}$ is divided into Algorithm $A_{21}$ and Algorithm $A_{22}$, Algorithm $A_{21}$ is further subdivided into two types that are abnormal in Case 1 but do not affect the status and new abnormalities in Case 2 but can be classified. In the case of Algorithm $A_{22}$, it is hoped that there are no rules to find a method through deep learning or different mechanical learning, and it can be classified back to $A_{21}$.

### 2.1. $A_{1}$ Detect Current Status

In the algorithm $A_{1}$, we give a threshold value, assuming the threshold value is $\theta_{I}=v$, the current C (Currentage) $I_{t}>\theta_{I}$. Then, it is judged to be an activated state. In addition, we record a mark $\Psi_{I}=1$ in the instrument, set $\phi_{tol}=\varepsilon$ if the current returns to $110-\phi_{tol}<C$$<110+\phi_{tol}$ Mark $\Psi_{I}=0$, and continue to monitor the normal current. We continue to monitor for a period of $\Gamma_{t}=K$ points. If the current is normal, we continue to monitor to maintain the operation of the machine, the calculation is referred to Algorithm 1.
**Algorithm 1:**$A_{1}$ Determine the fan current status**Require:** Set $\theta_{I}=v$, $\phi_{tol}=\varepsilon$  **if**
$V>\theta_{I}$
**then**    It is judged that the current is in the starting state. Set a flag $\Psi_{A_{1}}=1$    Continuously monitor for a period of time $\Gamma_{t}=K$ minutes.    **if** $C<110-\phi_{tol}\mathrm{or}C>110+\phi_{tol}$ **then**        Send a warning message, set $\Psi_{A_{1}}=0$ and continuously monitor the current.    **end if**  **end if**

### 2.2. $A_{2}$ Looking for Current Status and Heater

When the current exceeds the range, the system will send a warning message, and further enter the AI monitoring process through $A_{2}$, $A_{21}$ $A_{22}$, look for the corresponding monitoring process and determine whether it belongs to the historical classification situation. If it is a new situation, it will use the algorithm of $A_{22}$ Mechanism, combining manual and unsupervised learning with studying and judging the situation. The calculation is referred to Algorithm 2.

#### 2.2.1. $A_{21}$ Find Rule

Abnormal conditions occur, but do not affect the operation of the equipment, nor it will affect the machining process, so it belongs to the monitoring record, but it does not need to do any processing. The specific hardware assumes that X1,X2,… are not at the normal voltage but the system is functioning normally. If $\Psi_{A_{21}}=0$, then the monitoring light signal returns to the normal monitoring status. None of the above rules sets $\Psi_{A_{21}}=1$, executes $A_{22}$. Exceptions have rules given in Algorithm 3.
**Algorithm 2:**$A_{2}$ Looking for Current status**Require:** Set $\phi_{tol}=\varepsilon$,**Ensure:** Load AI package and monitoring module.  **if**
$110-\phi_{tol}<C<110+\phi_{tol}$
**then**    $\Psi_{A_{1}}=1$▹ indicating abnormal status  **else**    $\Psi_{A_{1}}=0$ to send a warning message▹ indicating abnormal status    It is judged that it belongs to $A_{21}$ with exceptions and rules $\Psi_{A_{21}}=1$    It is judged that it belongs to $A_{22}$ with exceptions and rules $\Psi_{A_{22}}=1$  **end if**

**Algorithm 3:**$A_{21}$ Find rule**Require:** The voltage of the device $X_{1}$, The voltage of the device $X_{2}$, *…*, The voltage of the device $X_{n}$. **if**
$C_{X_{1}}-\phi_{tol}<C<C_{X_{1}}+\phi_{tol}$
**then**  $\Psi_{A_{1}}=1$, $\Psi_{A_{21}}=0$ ▹ Normal standard **else if**
*…*
**then**  $\Psi_{A_{1}}=1$, $\Psi_{A_{21}}=0$  ▹ Normal standard **else if**
$C_{X_{n}}-\phi_{tol}<C<C_{X_{n}}+\phi_{tol}$
**then**  $\Psi_{A_{1}}=1$, $\Psi_{A_{21}}=0$▹ Normal standard **else**  $\Psi_{A_{1}}=0$, $\Psi_{A_{21}}=0$▹ Normal standard  **if** $\Psi_{A_{1}}=0$, $\Psi_{A_{21}}=0$ Conform to the rules **then**    Abnormal but does not affect the status. For exceptions, use data to determine the category, and find exception rules among the exceptions. If it does not match, look for a classification method.    $\Psi_{A_{1}}=1$, $\Psi_{A_{21}}=Ø$  **else**    Classify new anomalies. Use supervised or unsupervised learning, enhanced learning, use deep learning, new abnormal state, and at the same time conform to the abnormal state of historical data. To use classification methods, record them for later use or detection, and record the results.    Set up $\Psi_{A_{22}}=1$, Judgment belongs to $A_{22}$ There are no rules for exceptions, and unsupervised learning is used to detect whether there are new classifications.  **end if** **end if**

#### 2.2.2. $A_{22}$ No Rule Manual Confirmation

When $A_{22}=1$ is set as an exceptional case, we record the data, and the message is sent to the information Server side, and then, expert judgment is required to clarify the rule classification and analysis. Processing is complete once $\Psi_{A_{1}}=1$, $\Psi_{A_{22}}=Ø$, and for the calculation, we refer to Algorithm 4.

### 2.3. Classification Models

In this research, the normal or abnormal state of the system is examined through the classification of the state. Therefore, we use two machine learning methods to discuss the classification. One method is support vector machine (SVM) and the other is the most popular eXtreme Gradient Boosting (XGboost) method and is introduced as follows:
**Algorithm 4:**$A_{22}$ Manual confirmation**if**$\Psi_{A_{22}}=1$**then**    Expert analysis and record.    Use unsupervised learning to detect whether there are new classifications.    Set $\Psi_{A_{1}}=1$, $\Psi_{A_{22}}=Ø$ and confirm to return to the monitoring state to continuously monitor the current.**end if**

We divide anomaly detection into two parts, see Figure 2. Part 1 is a temperature monitoring and part 2 current monitoring. The system can use the time to query the electricity consumption statistics of a specific device at a specific time and provide data download [23,24,25].

Temperature abnormality monitoring:(a)Abnormal temperature: abnormal temperature of each part (Abnormal heating curve).(b)Find the fault: confirm the cause for repair.(c)Maintenance focused on fixing equipment after it broke.Current abnormality monitoring:Power consumption status of each component.Warning conditions: abnormal conditions, abnormal features, feature extraction.Display the cause of the failure: by capturing the *K*% of the current difference as a judging feature, we can establish an early warning rule to monitor the power consumption of the equipment during the overall process, and give an early warning when an abnormality occurs.Find the fault: confirm the cause for repair.

### 2.4. Support Vector Machine

SVM first appeared in 1963 and was proposed by Vladimir Vapnik and Alexey Chervonenkis. In 1992, a more powerful model was proposed by Bernhard Boser, Isabelle Guyon, and Vladimir Vapnik, which can create nonlinear classifiers by applying kernel techniques to the maximum margin hyperplane [26]. We want to find a classifier (a hyperplane) that can map $x_{i}$’s into higher-dimensional space so that the two different classes $y_{i}=1$ and $y_{i}=-1$ of points can be divided. The hyperplane can be expressed as the following formula:$$x_{i}w+b=0$$

We hope that the distance of the parallel hyperplane can separate the two groups of data as much as possible. The formula is as follows:(1)$$x_{i}w+b=+1$$
and
(2)$$x_{i}w+b=-1$$

We minimize ∥w∥ since it is always positive, *i*, $x_{i}$ and $y_{i}$, where *i* runs from 1 to *N*, *N* is the number of datasets. The following constraints apply: (3)$$\begin{array}{cc} x_{i}w+b\geq +1, & y_{i}=+1 \end{array}$$(4)$$\begin{array}{cc} x_{i}w+b\leq -1, & y_{i}=-1 \end{array}$$(5)$$\begin{array}{cc} y_{i}\left(x_{i}w+b\right)-1\geq 0, & \forall i \end{array}$$

From Equation (Equation 3) to (Equation 5), we can solve the following optimization problem:(6)$$\begin{array}{cc} \min_{w}\; & \frac{{∥w∥}^{2}}{2}\\ s.t.\; & y_{i}\left(x_{i}w+b\right)-1\geq 0 \end{array}$$
$y_{i}\left(x_{i}w+b\right)-1\geq 0$, for all i∈N. In mathematical optimization, this is a constrained optimization problem for finding the local maxima and minima of a function subject to equality constraints. Because it is quadratic, the surface is a paraboloid, with just a single global minimum. We can solve it by the Lagrangian method.

### 2.5. eXtreme Gradient Boosting

The XGBoost algorithm is classified as a regression method, so the output can be a continuous value. The idea is to select a few features to make a weak classification decision tree. At this time, the model outputs the predicted value and compares it with the actual value. The difference can also be said to be residual. As the new actual value of the data, we select the feature to train the next decision tree, so as to achieve the purpose of reviewing the error part of the previous step. Because the residual is used as the actual value to discuss, the predicted value calculated by the model must be added to the predicted value of the previous step, and so on [27,28,29].

Suppose the dataset is $X={\left(x_{1},x_{2},\ldots ,x_{n}\right)}^{tr}$. The corresponding value is $Y={\left(y_{1},y_{2},\ldots ,y_{n}\right)}^{tr}$. The superscript “tr” represents the transpose of the matrix. For the data, $x_{i}$ model predictive output can be written as a function
(7)$$\begin{array}{c} f_{t}\left(x_{i}\right)={\hat{y_{i}}}^{\left(t\right)} \end{array}$$
Among them, i=1,2,⋯,n, and “*t*” represents the model of the first step, and defines $f_{0}\left(x_{i}\right)=0.$ So, according to the above definition, the data $x_{i}$ go through each step. The total output of the model can be written as a general formula
(8)$$\begin{array}{c} {\hat{y_{i}}}^{\left(t\right)}={\hat{y_{i}}}^{\left(t-1\right)}+f_{t}\left(x_{i}\right) \end{array}$$
Therefore, the objective function of XGBoost is defined as a loss function plus a regular term used by the model to control the function to avoid overfitting $\Omega (f_{t})$.
(9)$$\begin{array}{cc} J^{\left(t\right)} & =\sum_{i=1}^{n}L\left(y_{i},{\hat{y_{i}}}^{\left(t\right)}\right)+\Omega \left(f_{t}\right)\\ \; & =\sum_{i=1}^{n}L\left(y_{i},{\hat{y_{i}}}^{\left(t-1\right)}+f_{t}\left(x_{i}\right)\right)+\Omega \left(f_{t}\right) \end{array}$$
There are many ways to choose the loss function *L*, such as using the mean square error (MSE), and the regular term is
(10)$$\begin{array}{cc} \Omega (f_{t}) & =\gamma T_{t}+\frac{1}{2}\lambda \sum_{j=1}^{T_{t}}w_{j}^{2} \end{array}$$
where γ,λ are arbitrary parameters, $T_{t}$ is the total number of leaf nodes of the model at step *t*, labeled 1,2,…,T; $w_{j}$ is the weight of the leaf node numbered *j*, that is, it is divided into a certain leaf node model that will output the value, $j=1,2,\ldots ,T_{t}$. Then, XGBoost uses a Taylor expansion to expand the loss function *L* to the second order so that the objective function can be written as
(11)$$\begin{array}{cc} J^{\left(t\right)} & \approx \sum_{i=1}^{n}L\left(y_{i},{\hat{y_{i}}}^{\left(t-1\right)}\right)+g_{i}f_{t}\left(x_{i}\right)+\frac{1}{2}h_{i}^{2}{\left(f_{t}\left(x_{i}\right)\right)}^{2}+\Omega \left(f_{t}\right) \end{array}$$
where $g_{i}=\frac{\partial L(y_{i},{\hat{y_{i}}}^{\left(t-1\right)})}{\partial {\hat{y_{i}}}^{\left(t-1\right)}}$ , $h_{i}=\frac{\partial^{2}L\left(y_{i},{\hat{y_{i}}}^{\left(t-1\right)}\right)}{\partial {\left({\hat{y_{i}}}^{\left(t-1\right)}\right)}^{2}}$, they are the first and second partial derivatives of $L(y_{i},{\hat{y_{i}}}^{\left(t-1\right)})$ to ${\hat{y_{i}}}^{\left(t-1\right)}$, respectively. The above process is to describe the XGBoost algorithm. Therefore, for this problem, we first calculate the continuous predicted value according to the original XGBoost and then use it in the previous part of the logistic regression. The Softmax function maps the value to a discrete probability value to solve this problem.

## 3. Current Monitoring System and Numerical Results

The goal of this system is to monitor the equipment and display the status of the equipment in real time by capturing the current value of each piece of equipment. After artificial intelligence technology is incorporated in the future, it is hoped that predictive maintenance can be achieved. Figure 3 displays the current monitoring system’s content, the cause of the abnormality, and the related status corresponding to the abnormal device. We use three kinds of warning information: normal (green light), abnormal (red light), and warning (yellow light) to represent the monitoring light of the system.

The hot-pressing furnace monitoring system uses three hot-pressing furnaces numbered 715, 930, and 1230. As shown in Figure 4, the equipment is divided into general heaters (HO1, HO2, HO3, etc.), fans, and SCR (Silicon Controlled Rectifier).

In Algorithm 1, $A_{1}$ determines the current status. In the data acquisition part, data from the three currents (R,S,T) of each device are retrieved every 30 min, and each device will have one dataset per minute, a total of 30 strokes in 30 min; so, we use the data of 30 strokes to judge the abnormality of the equipment.

The judgment logic is divided into two types: general heater and fan, and SCR (Silicon Controlled Rectifier). In Algorithm 2, $A_{2}$ is looking for the Current status and heater. In the abnormal judgment of the general heater and fan, we divide it into two algorithms, and use the warning score and Alarm score to record the result of this abnormal judgment.

At the beginning, we first calculate the average of the three current values as $\bar{X}$, and the average of the difference between the three currents as C. First algorithm: If the absolute value of C is more significant than 10% of $\bar{X}$ and less than 20% of $\bar{X}$, it means that this dataset is abnormal, and we set the warning score to 1. Second algorithm: If the absolute value of C is greater than 20% of $\bar{X}$, the dataset is abnormal. At this time, we set the Alarm score to δ (Algorithm 2).

$\bar{X}=\frac{R+S+T}{3}$.$Y=\frac{\left(RS)+(ST)+(RT\right)}{3}$.

In Algorithm 3, $A_{21}$ finds abnormal light signal judgment rule:**Green Light** δ<5, γ<10.**Yellow Light** δ<5, γ>10.**Pink Light** δ>5.**Red Light** A phase current of 715 A and the previous minute exceeds 200 A.At any one of the equivalent currents.The current exceeds 600 A between any equivalent current and the previous minute.

Here, δ is the alarm score and γ is the warning score. In the SCR abnormal judgment, we take the maximum value of the three currents (C) and the absolute value average of the three current differences (Z) for comparison, if

C=(max{R}+max{S}+max{T})/3.Z=(|(max{R}−max{S})|+|(max{S}−max{T})|+|(max{R}−max{T})|)/3.

Abnormal light signal judgment rule:**Green Light** C>0.9Z.**Yellow Light** 0.8Z<C<0.9Z.**Pink Light** C<0.8Z.**Red Light** 715: The difference from the previous minute when the R/S/T phase currents are either equivalent or exceed 200 A.930: When the difference between the R/S/T phase currents and the previous minute exceed 150 A at an equivalent level.

The results are displayed on each furnace’s electricity consumption record page, as shown in Figure 5. This page is divided into two parts. The left half is the equipment status of the hot-pressing furnace and the right half is the electricity meters, as shown in Figure 6, indicating which lights are displayed. It indicates the equipment status of the hot-pressing furnace, allowing warranty staff to check different equipment and speed up the process of equipment maintenance. The right side of the Figure shows the actual configuration of the meter and the monitoring equipment.

In addition, we can query the historical current record of the device on the right side of the device status, as shown in Figure 7. After selecting the time and device, we can know the three current values of the device. In addition, multiple devices can be selected at the same time to compare the values between the devices, as shown in Figure 8, so that the maintenance staff can know the status of the equipment before actually inspecting the equipment before further maintenance, which improves the efficiency of inspection.

The maintenance staff can further inquire about the current data of the equipment through this record to speed up the maintenance process. Figure 9 shows the abnormality record of the hot-pressing furnace.

This system also integrates the health status of each hot-pressing furnace for a whole day, as shown in Figure 10. The warranty staff can refer to these data to ensure whether the hot-pressing furnace has been in an abnormal state for a long time or the state has been exceeded. If the trend is getting worse, we repair the hot-press furnace in advance to achieve the effect of predicting the warranty.

In terms of calculation, the comprehensive risk light for equipment health refers to the result calculated from the previous abnormal records. If the green light is green, the score is 1, the yellow light is 2, the pink light is 3, and the red light is 3. Counting the score as 4, there will be a light record every thirty minutes, and we add up and average the scores throughout the day to get the health light for the day.

In the process of early warning judgment, this study proposes warning information by capturing the physical characteristics of the current and transmitting the information to the system to notify the relevant staff. Among them, the current data acquisition process is to query the electricity consumption statistics of a specific device through time through the platform’s application programming interface (API). The system provides the recording unit, and the record and data download information whose duration is the electricity consumption information, which can be extracted and obtained through the API. Figure 11 shows the visualization system, and the Single-Phase Current Abnormal Releases show the abnormal process. Figure 12 shows the alert State of the current monitoring system.

The operation process of the fan is divided into three steps. In the first step, the current is larger at startup. In the second step, the operation of the fan changes as the pressure and temperature increase. When the pressure is released, the operating current begins to decrease. In the third step, the process is over when the fan running current is 0. Figure 13 shows when the fan starts, Figure 14 shows when the process ends, and the fan stops.

To ensure the stable operation of the generator, the engine needs at least three windings. In the case of perfect balance, all three phases share the same load. Under normal conditions, the current difference between the three phases does not exceed 10% of the total (Figure 15, Three-phase load balancing). From Figure 16, it can be seen that the SCR current changes with time. Because the SCR mainly regulates the heating work, it will adjust the temperature as required. The current of the heater group will start with demand, and the process current is stable at about 60 A, as shown in Figure 17. This is the broken line diagram of the current when the fan is operating. It can be seen that there will be an increased starting current when the fan is started, and the production process will tend to be stable, as shown in Figure 18.

After the system is integrated into the artificial intelligence model, important information is understood from the historical electricity usage records and the field expert experience, which will be used as the judgment basis for the subsequent establishment of electricity usage early warning rules, and the predicted report will be issued. The steps are as follows: Step 1: Hot-pressing furnace actuation sequence. Step 2: Fan actuation process. Step 3: Current range when the heater is activated. Finally, three-phase load balancing.

The fan participates in the entire manufacturing process, and the starting current of 549 to 613 A keeps an average of 110 to 120 A during the operation process until the current is 0 A and the process ends. BANK 1 to 3 of the heater group, the current data maintain a constant value after the control is stabilized. When the heating wire fails, the currents of each phase range will fall between 53 and 55 or 60 and 63.

### Numerical Result

This subsection presents data and system implementation results and numerical results. Finally, the numerical results of the SVM and XGBoost classification models are presented.

We collect the current data of hot pressure, from 1 September 2020, to 31 October 2020, and obtain about 34 million data points, because 33.6 million of them have zero value or a very small current value. The ratio of the current value is about 6 to 4, so the number of small and zero values is deleted, and there are about 400,000 pieces of data that can be used for analysis. We obtain numerical results using SVM and XGBoost models and use 80% of the data as training data and the remaining 20% as test data. The validation profile uses 2519 data points from 0:00 to 1:59 on 11 March 2021, to verify the model’s accuracy. Finally, the accuracy of the two methods is compared using a confusion matrix and shown in the following Table 1.

Figure 19 shows the results of the hot-pressing furnace current monitoring feature extraction analysis. The data are automatically retrieved through API, and then, the pairwise difference between the three-phase currents and the average of the three currents are calculated as the training data, and the three load balances are set a label for each piece of data; 1 is abnormal, 0 is normal, and the maximum value of the three-phase current difference is greater than 1 to be considered or not, and less than 1 is considered normal. The orange box is the correlation between the label and the current difference. The darker the color, the higher the correlation. From a correlation point of view, the correlation of the current difference with LABEL is higher than that of the three-phase current.

In Table 2, the overall accuracy rate is 91.7%, the upper left is normal data, and the model prediction is normal. The upper right is normal data, but the model prediction is abnormal. The lower left is the abnormal data in the model prediction as normal, and the lower right is the abnormal data, and the model prediction is abnormal. The SVM model degree is stricter for normal data, and it is easy to misjudge normal as abnormal, so the accuracy rate is only 90% and XGBoost is as high as 97%. However, the confusion matrix of the XGBoost model predicts abnormal results under the same data. The XGBoost model has 11% misjudgments so that the abnormal conditions can be further improved through algorithms. Finally, we display the overall test set’s judgment data and the results’ prediction accuracy, shown in the table below.

The accuracy rate of XGBoost is as high as 99.96%. Other model results will also have high accuracy because system anomalies are incidental events.

## 4. Conclusions

In this research, artificial intelligence is applied to the predictive maintenance of the hot-pressing furnace. The abnormality and instability of the hot-pressing furnace are detected by monitoring current data combined with artificial intelligence. This research has developed a corresponding algorithm and artificial intelligence real-time system. Based on the actual situation, combined with the experience and technology of professional technicians and researchers, a corresponding algorithm and artificial intelligence real-time system have been developed, and real-time visual monitoring is presented, including algorithmic monitoring of abnormal and normal conditions. The monitoring system of this study can have several advantages, reducing huge downtime losses, reducing the occurrence of abnormal situations, strengthening the ability to predict, and the system can be continuously improved. The results of this system can continue to improve our algorithm, increase the accuracy of prediction, reduce the shutdown of abnormal conditions, and minimize economic losses. In the current monitoring, the model’s accuracy is as high as 0.89. XGBoost has the highest accuracy. Through the detection of abnormal effects and comparing the two models, XGBoost has a higher accuracy rate and a single type of error data. In constructing the artificial intelligence real-time system, we have established a hot-press furnace current data database, a health monitoring system, and an equipment abnormality warning push. Through such a system, it is hoped that the time for staff inspections can be reduced, the use of equipment will be continuously monitored online, and unplanned downtime will be reduced to zero. In the future, a three-phase current can be combined with fans and sensors to achieve predictive maintenance.

## Figures and Tables

**Figure 1 sensors-23-02230-f001:**
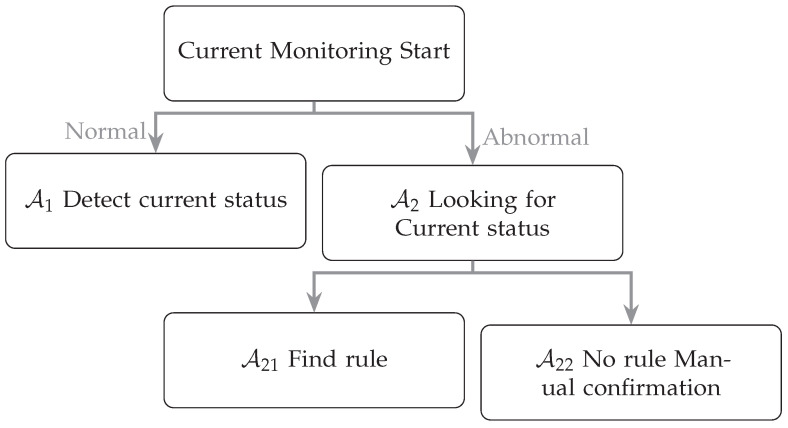
Algorithms for current and equipment status monitoring.

**Figure 2 sensors-23-02230-f002:**
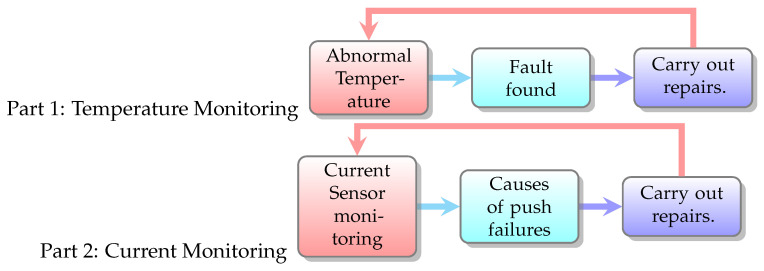
Cycle of Abnormal Detection.

**Figure 3 sensors-23-02230-f003:**
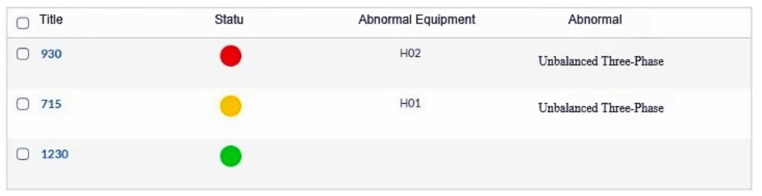
Current Monitoring System.

**Figure 4 sensors-23-02230-f004:**
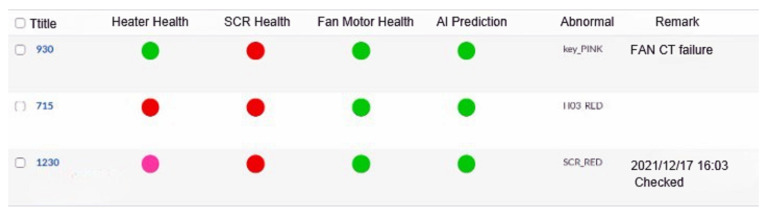
Hot-pressing furnace monitoring system.

**Figure 5 sensors-23-02230-f005:**
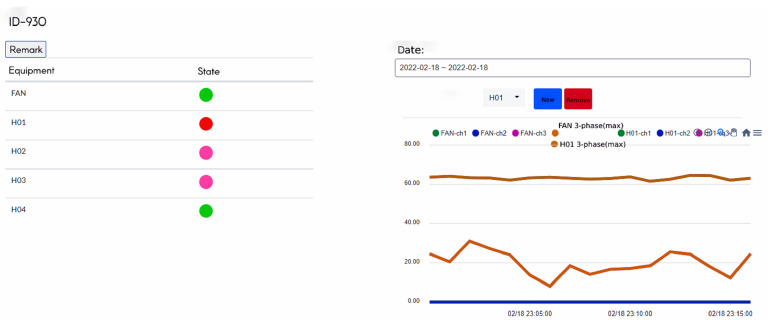
Hot-pressing furnace monitoring and electricity consumption record system.

**Figure 6 sensors-23-02230-f006:**
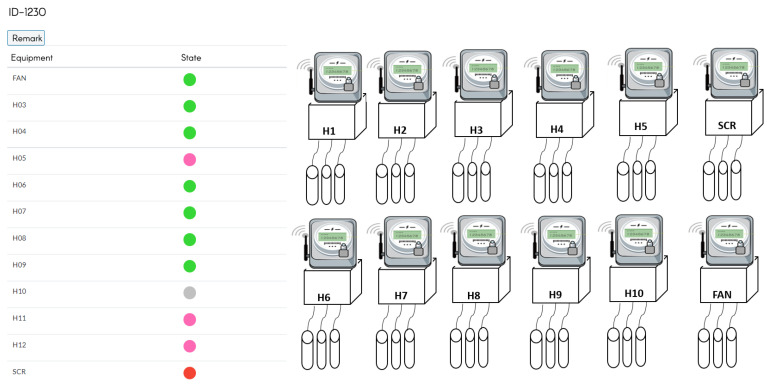
Hot-pressing furnace monitoring system and electricity meters.

**Figure 7 sensors-23-02230-f007:**
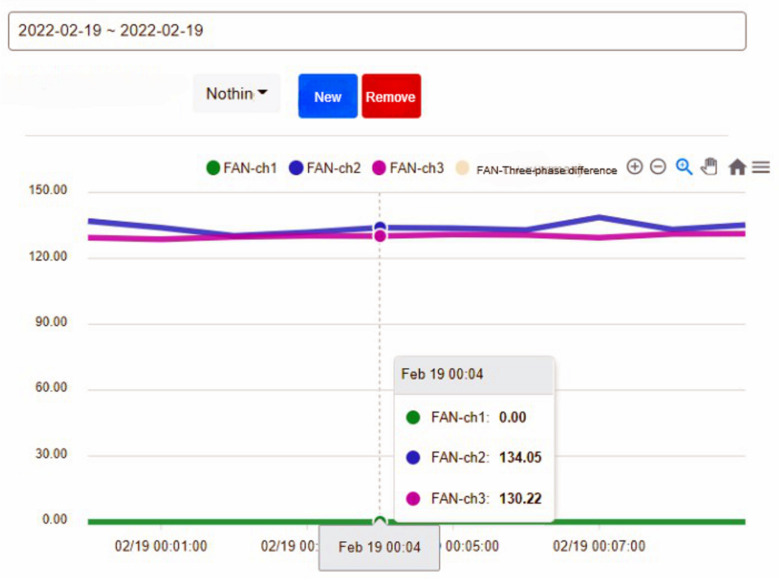
History of three current records.

**Figure 8 sensors-23-02230-f008:**
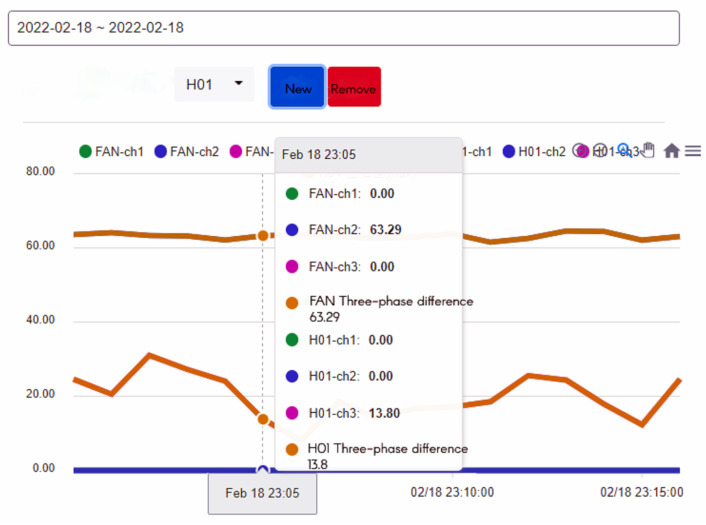
Comparison of devices, press furnace heat, equipment, and abnormal time values.

**Figure 9 sensors-23-02230-f009:**
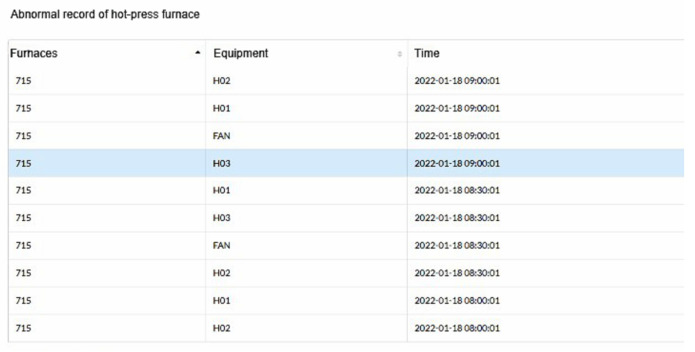
The abnormality record of the hot-pressing furnace.

**Figure 10 sensors-23-02230-f010:**
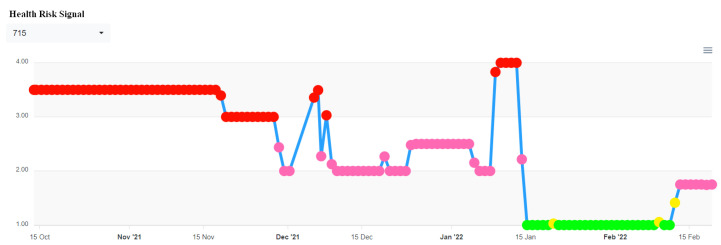
The abnormality record of the hot-pressing furnace.

**Figure 11 sensors-23-02230-f011:**
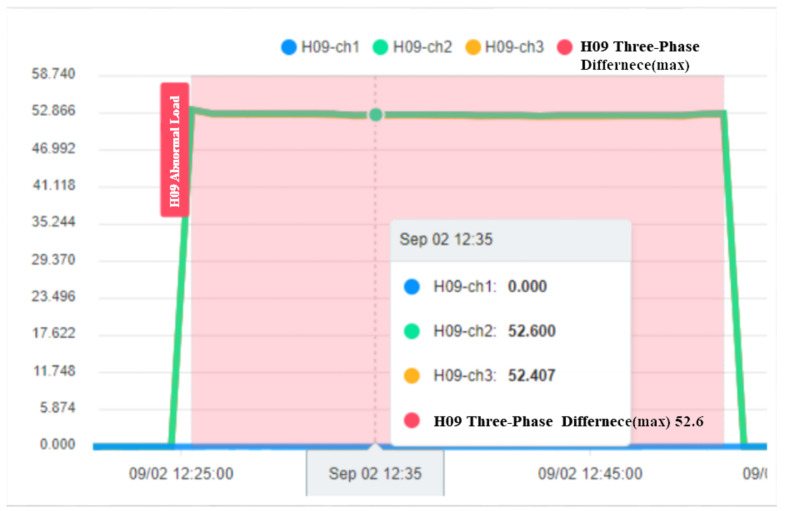
Visualization system.

**Figure 12 sensors-23-02230-f012:**
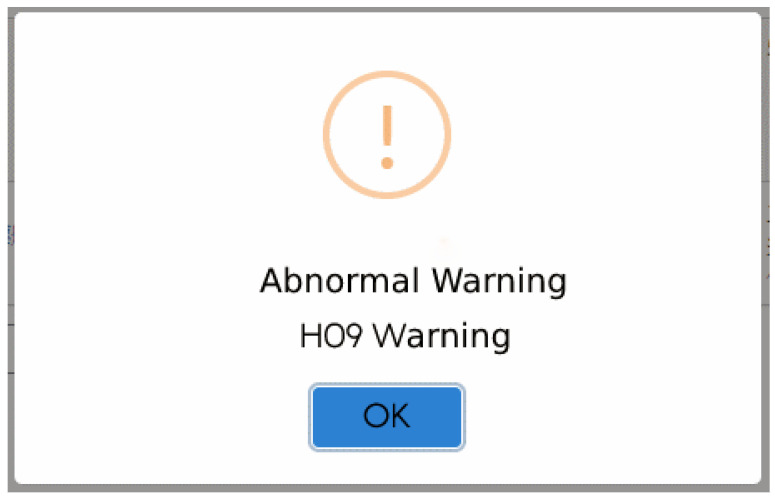
Abnormal Alarm.

**Figure 13 sensors-23-02230-f013:**
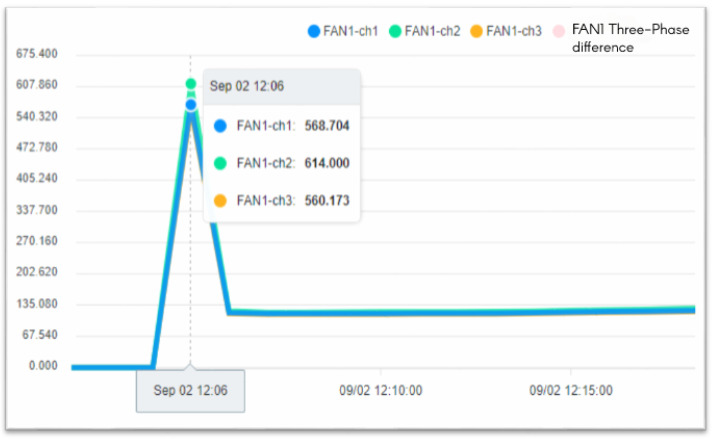
The System of Fan Start.

**Figure 14 sensors-23-02230-f014:**
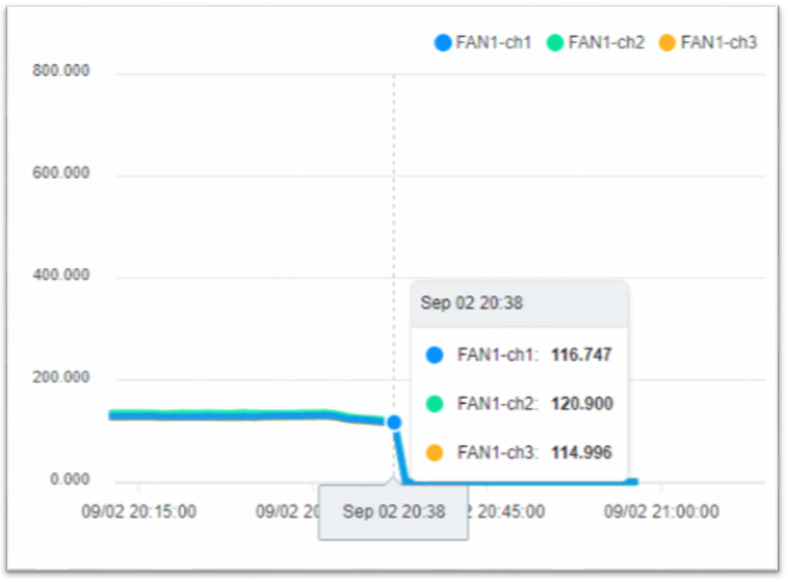
The System of Fan stops at the end of the process.

**Figure 15 sensors-23-02230-f015:**
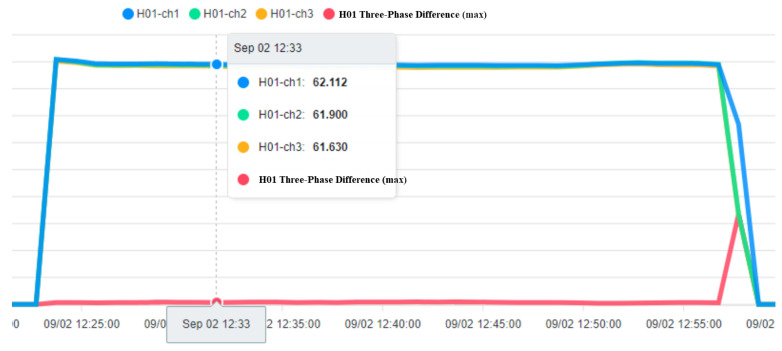
Balanced Three-Phase Load.

**Figure 16 sensors-23-02230-f016:**
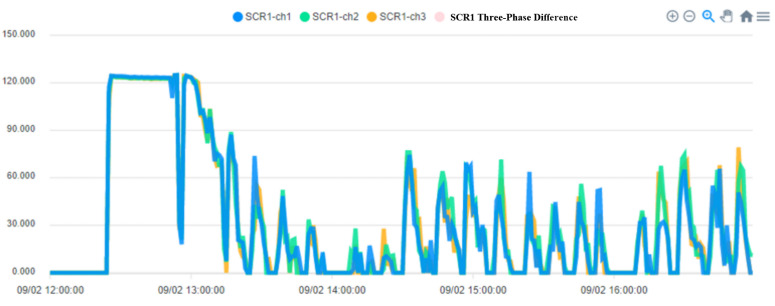
SCR Phased.

**Figure 17 sensors-23-02230-f017:**
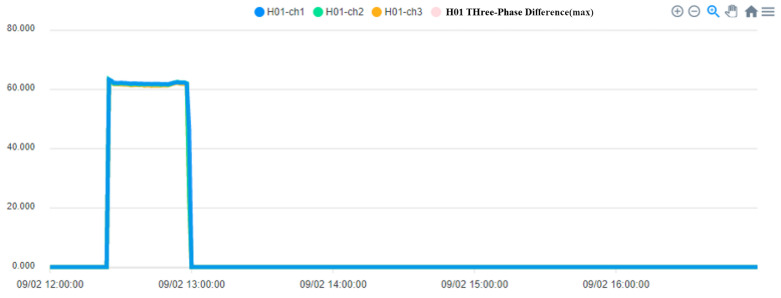
Heater System.

**Figure 18 sensors-23-02230-f018:**
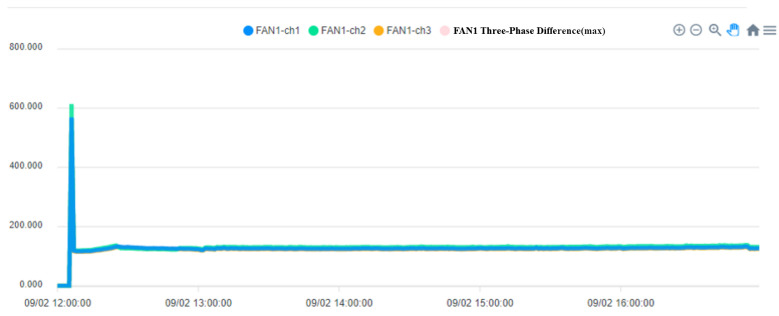
Fan Current system.

**Figure 19 sensors-23-02230-f019:**
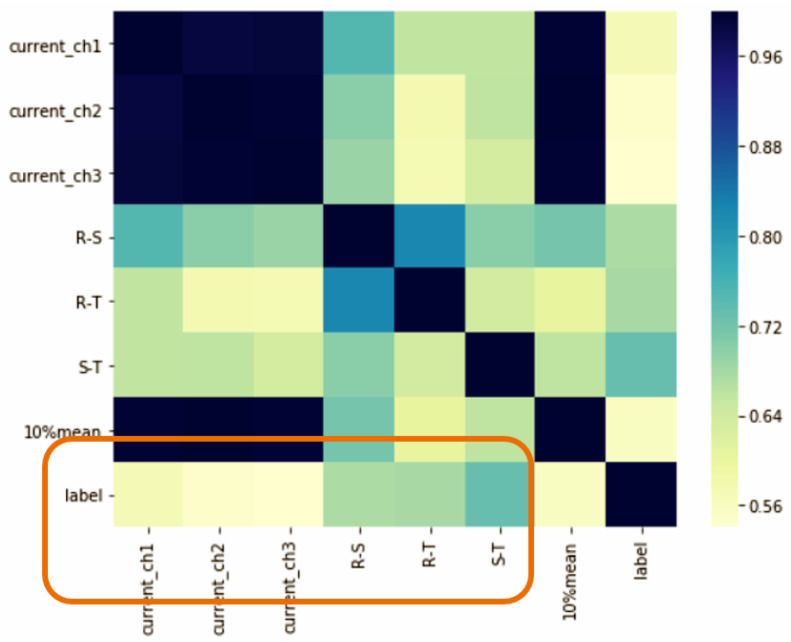
Feature Correlation.

**Table 1 sensors-23-02230-t001:** Confusion matrix of SVM and XGBoost model.

Confusion Matrix of SVM			Confusion Matrix of XGBoost		
**True/Predict**	**Normal**	**Abnormal**	**True/Predict**	**Normal**	**Abnormal**
Normal	0.90	0.1	Normal	0.97	0.03
Abnormal	0	1	Abnormal	0.11	0.89

**Table 2 sensors-23-02230-t002:** The accuracy of XGBoost and SVM model.

Model	Accuracy
SVM	0.9867
XGBoost	0.9996

## Data Availability

The data presented in this study are available on request from the corresponding author.

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
