# Peer review of "Hot-Pressing Furnace Current Monitoring and Predictive Maintenance System in Aerospace Applications"

_sensors, 2023, doi:10.3390/s23042230_

Round 1

Reviewer 1 Report

In this manuscript, artificial intelligence is applied to the predictive maintenance of the hot-pressing furnace. The abnormality and instability of the hot-pressing furnace are detected by monitoring current data combined with artificial intelligence algorithms. Generally, the submitted manuscript has some merits and is of practical significance for the predictive maintenance research field. The following comments can be considered to improve the quality.

1. Abstract

The abstract is a little logic mess. Please rewrite the abstract and clarify the concerned research topic in a clear and logic manner.

2. Introduction

Please clarify the research gaps and the corresponding motivations in a general reader-friendly manner. It is hard to quickly grasp the main research challenges of the concerned topic.

3. Introduction

The literature review is a little limited and not profound. The existing latest research achievements of the predictive maintenance system implementation issues under limited labeled samples and even no labeled samples (10.1016/j.ymssp.2022.109628), and the evident distribution discrepancies of the cross-domain prognosis should also be reviewed.

 4. Current monitoring system-Numerical result

The figures in the manuscript are a little vague, such as Fig. 3, Fig. 4, Fig. 5, Fig. 6, Fig. 7-Fig. 19. Please revise these figures and provide the high-quality PNG figures with clear legends and details in the figures.

 5. Current monitoring system-Numerical result

The accuracy of the proposed method is satisfactory. It is suggested to make more comparisons with the existing classification models to comprehensively verify the proposed method. The proposed method is only compared with SVM.

6. About the writing of this paper, grammar errors and typos need to be carefully checked and amended.

Author Response

Comments and Suggestions for Authors

In this manuscript, artificial intelligence is applied to the predictive maintenance of the hot-pressing furnace. The abnormality and instability of the hot-pressing furnace are detected by monitoring current data combined with artificial intelligence algorithms. Generally, the submitted manuscript has some merits and is of practical significance for the predictive maintenance research field. The following comments can be considered to improve the quality.

  1. Abstract

The abstract is a little logic mess. Please rewrite the abstract and clarify the concerned research topic in a clear and logic manner.

Answer:

The abstract has been re-corrected.

This research combines the application of artificial intelligence in the production equipment fault monitoring of aerospace components. It detects three-phase current abnormalities in large autoclave through smart meters and provides early preventive maintenance.  Different anomalies are classified and a suitable monitoring process algorithm is proposed to improve the overall monitoring quality and accuracy and monitoring stability by applying AI. We also designed a system to present the power consumption of the heater and fan of the autoclave and to visualize the process. Combining artificial intelligence with the experience and technology of professional technicians and researchers to detect and proactively grasp the health of the autoclave equipment improves the shortcomings of previous expert systems, achieves long-term stability and reduces costs. The complete algorithm introduces a model that corresponds to the actual production environment, with the best model result being XGBoost with an accuracy of 0.97.

  1. Introduction

Please clarify the research gaps and the corresponding motivations in a general reader-friendly manner. It is hard to quickly grasp the main research challenges of the concerned topic.

Answer:

The introduction has been re-corrected.

  1. Introduction

The literature review is a little limited and not profound. The existing latest research achievements of the predictive maintenance system implementation issues under limited labeled samples and even no labeled samples (10.1016/j.ymssp.2022.109628), and the evident distribution discrepancies of the cross-domain prognosis should also be reviewed.

Answer:

There has been added a new literature review.

  1. Current monitoring system-Numerical result

The figures in the manuscript are a little vague, such as Fig. 3, Fig. 4, Fig. 5, Fig. 6, Fig. 7-Fig. 19. Please revise these figures and provide the high-quality PNG figures with clear legends and details in the figures.

Answer:

There have been adjustments to the clarity of the graph.

  1. Current monitoring system-Numerical result

The accuracy of the proposed method is satisfactory. It is suggested to make more comparisons with the existing classification models to comprehensively verify the proposed method. The proposed method is only compared with SVM.

Answer:

These two models are highly accurate in most data science fields, while other models are close to or too different from these two models, thanks to the kind suggestions of the reviewers.

  1. About the writing of this paper, grammar errors and typos need to be carefully checked and amended.

Answer:

We have fixed the errors and adjusted the text, thanks to the kind suggestions of the reviewers.

Reviewer 2 Report

This is an interesting article on “Hot-Pressing Furnace Current Monitoring and Predictive Maintenance System in Aerospace Applications”, but it is necessary to rewrite the paper before to being accepted. I think the paper needs a major revision.

Comments and Suggestions

1- The research design need to be improved. The paper need to be rewritten focus in this structure.

·         Introduction

·         Material and methods

·         Results and discussion

·         Conclusions

2- Please check the different parts of the article at the end of the introduction, are they correct?

3- - The quality of the imagines need to be improved, sometime it is impossible to see the letters.

4- Please review the bibliography in the paper, sometime the place where they are placed, may not be appropriate.

5-  Please, be explicit and defining the main aim and scientific contribution of the study in the introduction.

6- P1, L22-29 Please add a reference at the end of the paragraph.

7- Check the English grammar throughout the paper, sometimes there are some misplaced dots and capital letters.

8- Please, review all the abbreviations throughout the paper, there are many without defining what they mean. Example: SVR, KNN, PSO, SCR, API, etc. …….

9-Please check P9, L237-243, change “personnel” for “Staff”, it looks better.

10- Please check P10, L268-269, “10Three-phaseload balancing”, it is right?

11- Please check P10, L293small The

Author Response

Comments and Suggestions

1- The research design need to be improved. The paper need to be rewritten focus in this structure.

  • Section 1 Introduction
  • Section 2 Material and methods
  • Section 3 Current Monitoring System and Numerical Result
  • Section 4 Conclusions

Answer:

We have fixed it, thanks to the kind suggestions of the reviewers.

  • Section 1 Introduction
  • Section 2 Monitoring with Artificial Intelligence and Algorithm Description
  • Section 3 Current Monitoring Ssystem and Numerical Result
  • Section 4 Conclusions

2- Please check the different parts of the article at the end of the introduction, are they correct?

Answer:

We have fixed it, thanks to the kind suggestions of the reviewers.

Line 77: The first section is the introduction, the second section is monitoring with artificial intelligence and algorithm description, the third section is the current monitoring system and numerical result, and the last section is the conclusion.

3- The quality of the imagines need to be improved, sometime it is impossible to see the letters.

Answer:

There have been adjustments to the clarity of the graph, thanks to the kind suggestions of the reviewers.

4- Please review the bibliography in the paper, sometime the place where they are placed, may not be appropriate.

Answer:

We have fixed it, thanks to the kind suggestions of the reviewers.

5-  Please, be explicit and defining the main aim and scientific contribution of the study in the introduction.

Answer:

We have fixed it, thanks to the kind suggestions of the reviewers.

Line 14:

This research mainly uses artificial intelligence to monitor equipment maintenance time and equipment health, adjust production schedules and purchase maintenance parts in advance, and reduce production costs due to downtime.

6- P1, L22-29 Please add a reference at the end of the paragraph.

add

7- Check the English grammar throughout the paper, sometimes there are some misplaced dots and capital letters.

Answer:

We have fixed it, thanks to the kind suggestions of the reviewers.

8- Please, review all the abbreviations throughout the paper, there are many without defining what they mean. Example: SVR, KNN, PSO, SCR, API, etc. …….

Answer:

We have fixed it, thanks to the kind suggestions of the reviewers.

Line 51: SVR (Support Vector Regression) model, a KNN (K-Nearest Neighbor) model

Line 63: SVM: performance of support vector machine (SVM).

Line 65: PSO, optimization particle swarm optimization (PSO) Algorithms.

Line 189:SCR,  (Silicon Controlled Rectifier )    

Line 258: application programming interface (API)

9-Please check P9, L237-243, change “personnel” for “Staff”, it looks better.

Answer:

We have fixed it, thanks to the kind suggestions of the reviewers.

Line 27: If an abnormal situation occurs in each device, it must be tested by maintenance staff,

Line 233: allowing warranty staff to check

Line 240: maintenance staff can know the status of the equipment

Line 242: The maintenance staff can further inquire
Line 246:
The warranty staff can refer to this data

Line 257: the system to notify the relevant staff

Line 342: it is hoped that the time for staff

10-Please check P10, L268-269, “10Three-phaseload balancing”, it is right?

Answer:

We have fixed it, thanks to the kind suggestions of the reviewers.

Line 271:10% of the total.

11-Please check P10, L293 “small The”

Answer:

We have fixed it, thanks to the kind suggestions of the reviewers.

Line 296: small. The

Round 2

Reviewer 1 Report

Thanks a lot for the efforts made by the authors. The revised manuscript has improved a lot. I don't have further questions.

Author Response

According to the reviewer’s comments, we have revised the manuscript extensively. If there are any other modifications we could make, we would like very much to modify them and we really appreciate your help. We hope that our manuscript could be considered for publication in your journal. Thank you very much for your help.

Reviewer 2 Report

This is an interesting article on “Hot-Pressing Furnace Current Monitoring and Predictive Maintenance System in Aerospace Applications”, but it is necessary to rewrite the paper before to being accepted. I think the paper needs a minor revision.

Comments and Suggestions

Please check the different parts of the article at the end of the introduction. Can you add this sentence to the beginning of the paragraph?

“The paper is arranged into four different sections”

Please review once more the bibliography in the paper, in some occasions it puts it like this [12] . [13] in others like [22,23], use the second one.

Check the English grammar throughout the paper, sometimes there are some misplaced dots.

Author Response

 Comments and Suggestions

Please check the different parts of the article at the end of the introduction. Can you add this sentence to the beginning of the paragraph?

“The paper is arranged into four different sections”

Answer:

We have added the sentence, thanks to the kind suggestions of the reviewers.

Line 77: The paper is arranged into four different sections. The first section is the introduction, the second section is monitoring with artificial intelligence and algorithm description, the third section is the current monitoring system and numerical result, and the last section is the conclusion.

Please review once more the bibliography in the paper, in some occasions it puts it like this [12] . [13] in others like [22,23], use the second one.

Answer:

Thank you again for your positive comments and valuable suggestions to improve the quality of our manuscript.

Line 36: Sacco et al. [4] uses ML vision systems to develop machine learning applications in composites

Line 45: Xiang et al. [5] identify abnormal samples in distribution transformer inspection

Line 46: Balouji et al. [ 6 ] integrates an advanced generative adversarial network to detect anomal

Line 53: Hadi Salih et al.[ 11 ] proposed using the Discrete Cosine Tran

Line 57: Zeng et al. use the parallel symmetric multiprocessor computing machine for problems with high-dimensional training sets. This new technique has significant advantages in speed without reducing the generalization performance of the support vector machine (SVM). Wang et al. improved the effectiveness of real power system experimental with parameters optimization particle swarm optimization (PSO) algorithms [13].

Line 64: Recently, industrial power systems [16 – 19] deployed the power consumption of 16 servers in this pilot implementation for a medium-sized enterprise.

Line 71: Zhu et al. combines Active learning (AL) and transfer learning (TL) for remaining useful life (RUL)  prediction method more practical with a lesser demand on the run-to-failure data under limited labeled samples and even no labeled samples [20].

Check the English grammar throughout the paper, sometimes there are some misplaced dots.

Answer:

Thanks. We have corrected these mistakes based on your suggestions.
